# Brain MRI Biomarkers in Isolated Rapid Eye Movement Sleep Behavior Disorder: Where Are We? A Systematic Review

**DOI:** 10.3390/brainsci13101398

**Published:** 2023-09-30

**Authors:** Stephan Grimaldi, Maxime Guye, Marta Bianciardi, Alexandre Eusebio

**Affiliations:** 1Department of Neurology and Movement Disorders, APHM, Hôpital Universitaire Timone, 265 rue Saint-Pierre, 13005 Marseille, France; 2Centre d’Exploration Métabolique par Résonnance Magnétique, Assistance Publique des Hôpitaux de Marseille, Hôpital Universitaire Timone, 265 rue Saint-Pierre, 13005 Marseille, France; 3Center for Magnetic Resonance in Biology and Medicine, Aix Marseille University, Centre National de la Recherche Scientifique, 27 Bd Jean Moulin, 13385 Marseille, France; 4Department of Radiology, Athinoula A. Martinos Center for Biomedical Imaging, Massachusetts General Hospital and Harvard Medical School, 149 13th St., Charlestown, MA 02129, USA; 5Division of Sleep Medicine, Harvard University, Boston, MA 02114, USA; 6Institut de Neurosciences de la Timone, Aix Marseille University, Centre National de la Recherche Scientifique, 27 Bd Jean Moulin, 13385 Marseille, France

**Keywords:** biomarkers, connectivity, functional brain imaging, iron, neurodegenerative disorders, neuroimaging, neuromelanin, Parkinson’s disease, REM sleep behavior disorder, ultra-high field MRI

## Abstract

The increasing number of MRI studies focused on prodromal Parkinson’s Disease (PD) demonstrates a strong interest in identifying early biomarkers capable of monitoring neurodegeneration. In this systematic review, we present the latest information regarding the most promising MRI markers of neurodegeneration in relation to the most specific prodromal symptoms of PD, namely isolated rapid eye movement (REM) sleep behavior disorder (iRBD). We reviewed structural, diffusion, functional, iron-sensitive, neuro-melanin-sensitive MRI, and proton magnetic resonance spectroscopy studies conducted between 2000 and 2023, which yielded a total of 77 relevant papers. Among these markers, iron and neuromelanin emerged as the most robust and promising indicators for early neurodegenerative processes in iRBD. Atrophy was observed in several regions, including the frontal and temporal cortices, limbic cortices, and basal ganglia, suggesting that neurodegenerative processes had been underway for some time. Diffusion and functional MRI produced heterogeneous yet intriguing results. Additionally, reduced glymphatic clearance function was reported. Technological advancements, such as the development of ultra-high field MRI, have enabled the exploration of minute anatomical structures and the detection of previously undetectable anomalies. The race to achieve early detection of neurodegeneration is well underway.

## 1. Introduction

Despite tremendous improvement in the diagnosis and management of patients with Parkinson’s disease (PD), this disease remains incurable. In an attempt to limit the progression of neurodegeneration, many innovative treatments are being developed and tested, such as immunotherapy, gene therapy, and drug repurposing [1]. However, these therapies can only be offered to PD patients with motor symptoms because clinical examination is the only reliable marker able to assess their efficacy.

It is therefore urgent to develop early biomarkers able to identify neurodegeneration before the onset of motor symptoms and offer premotor patients a dedicated treatment. Moreover, these biomarkers are expected to reflect the stability or worsening of neurodegeneration during the trial of innovative treatments.

The onset of motor symptoms occurs many years after the commencement of the neurodegenerative process and about 30% of nigrostriatal synaptic activity is already lost [2]. During the premotor phase, several clinical symptoms are associated with a significant risk of developing in the following years a synucleinopathy (a group of neurodegenerative diseases including PD, dementia with Lewy bodies (DLB), and multiple system atrophy (MSA)). These include, in particular, isolated rapid eye movement (REM) sleep behavior disorder (iRBD), hyposmia, and constipation [3]. Instead, late-onset depression is classified as a low-risk factor for prodromal PD with low sensitivity and specificity because of the lack of dedicated studies [3]. 

Brain imaging showed promising results toward the goal of identifying early biomarkers of neurodegeneration in high-risk patients. For example, positron emission tomography (PET)/single-photon emission computed tomography (SPECT) imaging of striatal membrane dopamine transporters (DAT) revealed a dopaminergic deficit in approximately 50% of patients with idiopathic/isolated RBD (iRBD) [4]. Moreover, leucine-rich kinase 2 (LRRK2) non-manifesting carriers had reduced DAT binding compared with noncarriers [5]. Alternatively, metaiodobenzylguanidine (MIBG) myocardial scintigraphy in patients having at least two symptoms among constipation/iRBD/orthostatic hypotension, revealed abnormal results in 94% of cases [4]. Nevertheless, each of these techniques is very specific to a nervous system pathway and cannot evaluate different parameters at the same time (in the present example, it was the nigrostriatal dysfunction or the cardiac sympathetic nervous system). Furthermore, their availability is sometimes still difficult for the general population. As an alternative, brain MRI is potentially a multimodal technique to investigate prodromal PD. Interestingly, technical development of brain MRI in recent years, made it possible to explore very small anatomical structures and to reveal previously undetectable anomalies, especially in PD [6].

To identify promising biomarkers reflecting neurodegeneration in the premotor stage of synucleinopathies, we conducted a systematic review of the literature regarding MRI abnormalities at prodromal disease stages, especially in patients without any sign of parkinsonism but having iRBD.

## 2. Materials and Methods

To achieve this aim, we adopted the relevant criteria of the preferred reporting items for systematic reviews and meta-analyses (PRISMA) [7]. In June 2023, we conducted a literature search via the MEDLINE (PubMed) and Google Scholar databases of articles related to structural and functional MRI studies in prodromal PD. The following MRI modalities were considered for the review: structural, diffusion, functional, iron-sensitive, neuromelanin-sensitive MRI, and proton magnetic resonance spectroscopy. We used the combination of the following terms: “magnetic resonance imaging (MRI)” + “prodromal PD” or “premotor PD” or “preclinical PD” or “prodromal synucleinopathy” or “premotor synucleinopathy” or “preclinical synucleinopathy” or “non-motor” or “idiopathic RBD” or “isolated RBD” + “Parkinson” or “voxel based morphometry” or “cortical thickness” or “subcortical volume” or “relaxometry” or “iron” or “Quantitative Susceptibility Mapping” or “QSM” or “susceptibility weighted imaging” or “SWI” or “T2*” or “R2*” or “neuromelanin” or “diffusion tensor imaging” or “spectroscopy” or and “functional MRI” filtered for the period from 2000 to 2023. After removing duplicates, as well as screening the titles, abstracts, and/or full texts for suitability, we selected 77 studies and excluded 5169 studies. Details of the steps of the database search are given in Figure 1. 

Specifically, the following criteria were applied for inclusion in the review: (1) English full text available; (2) original articles or case–control studies that included patients with iRBD; (3) iRBD diagnosis confirmed by video-polysomnography; (4) MRI studies, and (5) absence of motor sign of parkinsonism or not meeting the criteria for PD diagnosis.

To reduce the risk of omitting a report, the search was conducted twice at 6-month intervals and the bibliography was carefully evaluated by all authors. This study was not registered in PROSPERO (International prospective register of systematic reviews).

Section A.1 briefly describes the different MRI modalities used in the studies we report. Readers can refer to it for a better understanding of the image processing methods used.

## 3. Results

### 3.1. iRBD as a Prodromal Marker of Synucleinopathy

According to the International Classification of Sleep Disorders 3rd Edition, rapid eye movement (REM) sleep behavior disorder (RBD) is a parasomnia manifested by vivid, often frightening dreams accompanied by simple or complex motor behaviors during REM sleep [8]. The diagnosis is confirmed with polysomnography. RBD presents with a persistent muscle tone during REM sleep, resulting in either sustained (tonic) excessive activity during REM sleep in the chin EMG, intermittent (phasic) excessive activity during REM sleep in the chin or limb EMG, or both [8]. Idiopathic (or isolated) RBD (iRBD) is an established early manifestation of neurodegenerative disease, i.e., synucleinopathy [9]. Indeed, in 174 patients diagnosed with video polysomnography-confirmed iRBD, the estimated risk of a neurodegenerative syndrome was 33.1% five years after the diagnosis of iRBD, 75.7% after 10 years, and 90.9% after 14 years [10]. Moreover, reduced striatal DAT signaling was reported in patients with iRBD, with gradually increasing loss of tracer uptake across the continuum from iRBD to PD [11]. Nevertheless, it is important to know that dream enactment behavior may also be observed in narcolepsy [12], autoimmune diseases (including IgLON5-antibody), or brainstem lesions [13], as well as with certain medications, mostly selective serotonin or serotonin-norepinephrine reuptake inhibitors [14]. Degeneration of the circuits of the nucleus subcoeruleus, which projects to the medullary reticular formation and spinal ventral horn interneurons, releases the inhibition of spinal motor neurons that results in characteristic motor behaviors [13]. There are a few arguments supporting the involvement of the motor cortex in the origin of the violent movements during iRBD. First, some premotor neurons located in the pontomedullary reticular nuclei receive projections from pyramidal neurons in the motor cortex [15]. Second, prolonged and elaborated complex motor behaviors (such as drawing or singing) also support the role of the motor cortex in iRBD [16]. Finally, ictal SPECT during RBD episodes in iRBD patients showed that the supplementary motor area (SMA) was the generator of these behaviors possibly bypassing the basal ganglia [17]. Moderate-to-severe neuronal loss, gliosis, Lewy bodies, and Lewy nerites were found in the subcoeruleus nucleus, the gigantocellular reticular nucleus, and the pedunculopontine nucleus in patients with iRBD and antemortem diagnosis of PD and DLB [18]. Autopsy in patients with iRBD is very rare. One case reported the presence of Lewy body disease with a marked decrease of pigmented neurons in the locus coeruleus and Substantia Nigra (SN) [19], whereas the other one reported Lewy bodies, Lewy neurites, and neuronal loss that were present in the dorsal motor nucleus of the vagus and the medullary tegmentum, including the region encompassing the gigantocellular reticular formation (GCRF). The locus coeruleus had no significant neuronal loss and gliosis but did have many Lewy bodies and Lewy neurites [20]. More recently, a large multicentric study found that genes involved in mitochondrial function and macroautophagy were the strongest contributors to cortical thinning occurring in iRBD [21].

### 3.2. Key MRI Features in iRBD

iRBD is undoubtedly the most studied premotor manifestation of synucleinopathies. The most important MRI features found in iRBD patients are listed in Table 1. 

In summary, in iRBD there was a volumetric reduction in the striatum and lenticular nucleus [22,23] along with decreased volume or thickness mainly in the frontal [22,24,25,26,27,28,29] cortex but it was also reported in the posterior temporal, inferior parietal, dorsolateral prefrontal and lateral occipital cortices [29].

In patients with iRBD, a brain-clinical pattern in patients was found and predicted conversion to Dementia with Lewy Body (DLB) [30,31].

There was a consistent loss of neuromelanin in the SN [32,33] and locus coeruleus [34,35] while iron accumulation in SN, and more precisely in nigrosome 1 [36], was also reported [37,38,39].

Functional network iRBD studies showed heterogeneous results. The connectivity within the basal ganglia was mostly decreased [40,41,42,43] yet this finding was associated with increased connectivity with other regions participating in the coordination of movement and its execution (cerebellum, premotor cortex, SMA) suggesting a compensatory mechanism [44] without any concrete evidence to this day.

Finally, MRS results on iRBD showed changes in neuron function and metabolism [45,46], yet more investigations have to be conducted at this stage of the disease. 

As a result, measures of atrophy, iron, and NM-sensitive MRI might be the most regular and promising features for an early neurodegenerative diagnosis. 

**Table 1 brainsci-13-01398-t001:** Key MRI features in idiopathic rapid eye movement (REM)-sleep behavior disorder (iRBD) revealed in studies utilizing advanced post-processing algorithms and large subject cohorts.

MRI Modalities	Abnormalities
**Structural MRI**	Decreased volume in the frontal cortex [21,25], dorsolateral prefrontal, occipital cortices [47,48]
	At baseline, a Dementia with Lewy Body-pattern was significantly higher in dementia-first than in parkinsonism-first converters [30,31,47]: thinning of the temporal, orbitofrontal, and insular cortices and relative preservation of the precentral and inferior parietal cortices
	iRBD patients with MCI showed a pattern of local deformation and volume atrophy in the cortical and subcortical (regions compared to those without MCI or controls. These findings were linked to diminished performance in attention and executive functions, visuospatial abilities, and increased severity of motor symptoms [49].
	Cortical thinning in the right visual area V4 in patients with iRBD was coupled with impaired pursuit initiation [48]
	Some studies reported an increase in volume in both hippocampi [50], suggesting a compensatory mechanism
**Diffusion MRI**	Heterogeneous results: Decreased FA in the SN [32], rostral pons [50], fornix and right visual stream (V1,V2), internal capsule bilaterally [51]
	Decreased MD and axial diffusivity in SN and increased MD within the pontine reticular formation [50] Increased MD in the right corticospinal tract [52].
	Increased mean free-water values (bi-tensor DTI) in the posterior SN [53]
	Impaired structural connectivity between REM-on (fe.g., subcoeruleus) and REM sleep muscle atonia (e.g., medullary reticular formation) areas [54]
**Iron-sensitive MRI**	Less than 2/3 of subjects had a loss of DNH (nigrosome 1) in the SWI sequence, [37,55,56,57]
	No difference in SN [32] concerning R2*
	Studies analyzed iron overload, with higher values in SN [39], right caudate, right/left dentate, and left caudate nuclei [57] and no difference for the other studies [36,58]
**Neuromelanin-sensitive MRI**	More than 80% of subjects: reduced neuromelanin-sensitive volume and/or signal intensity in SN [33] and locus coeruleus [34,35]
**Proton Magnetic Resonance Spectroscopy**	Possible reduction of Choline/Creatine and N-acetylaspartate/Choline ratios in the pontine tegmentum [46]
**fMRI**	rsfMRI: Reduced nigro-striato-thalamo-pallidal inter- and intrahemispheric connectivity [40,41,42,43]
	rsfMRI: Elevated connectivity between SN and the right cuneus/precuneus/superior occipital [42]
	Cerebral blood flow and microvascular flow were disturbed in frontal cortical areas [59].
	Diffusion tensor image analysis along the perivascular space (DTI-ALPS) index showed lower values in iRBD indicating a reduction in perivascular diffusivity [60,61]
	In 4 out of 5 iRBD patients who developed MSA, MRI showed one of these signs: putaminal rim, cerebellar atrophy, middle cerebellar peduncle atrophy, middle cerebellar peduncle hyperintensity, pontine atrophy or cross-bun sign [62]

### 3.3. Grey Matter (GM) Integrity 

Different methods were used to evaluate GM integrity such as voxel-based morphometry (VBM), deformation-based morphometry (DBM), cortical thickness, and segmentations.

With regard to cortical GM, most of the studies reported decreased volume or thinning (Figure 2) in the frontal cortex [21,22,24,25,26,27,28,29] in iRBD compared to healthy subjects. Also, the posterior temporal, parietal [29,47], dorsolateral prefrontal, orbitofrontal, occipital cortices [29,47,48] the superior frontal sulcus [25], the right superior frontal sulcus [27], and the dorsolateral primary motor cortex in the right hemisphere [22] were thinner or with a reduced volume.

More marginally, other regions were reported to be affected: anterior cingulate [22,28,63] temporal [24,28], parietal [24], occipital cortices [24,28], the right and left cerebellum, tegmental portion of the pons, left parahippocampal gyrus [26], right posterior hippocampal [27], insula [63], postcentral, left superior parietal, and in lateral occipital regions [27], the lingual gyrus [25] and fusiform regions too [25,27].

To better understand the clinical relevance of these findings, some authors evaluated whether GM damage was correlated with clinical symptoms. The progression of motor signs in iRBD was associated with cortical thinning primarily in frontal regions [64]. Lower GM volume in the frontal lobes was associated with longer iRBD duration and lower age of iRBD symptoms onset [22]. Slower finger tapping of the right hand was associated with cortical thinning in the right paracentral and superior parietal lobule cortices, as well as in the right postcentral and superior parietal lobule cortices [22]. In 182 iRBD patients, thinning of the bilateral sensorimotor cortex and reductions in the surface area of frontopolar, sensorimotor, occipital, inferior parietal, lingual, and fusiform cortices were associated with motor impairment, whereas cortical thinning in bilateral insula, right temporal cortex, and left posterior temporal cortex were associated with cognitive impairment [29]. Parieto-occipital and orbitofrontal thinning were associated with visuospatial loss (Visual form discrimination test) [64]. 

Patients with iRBD and Mild Cognitive Impairment (MCI) had cortical thinning in the frontal, cingulate, temporal, and occipital cortices, and abnormal surface contraction in the lenticular nucleus and thalamus; whereas, those without any cognitive symptoms had cortical thinning restricted to the frontal cortex [28]. No neuropsychological data were correlated with hippocampal atrophy in patients with iRBD in another study [27]. 

Using DBM, individuals with iRBD and MCI exhibited a distinct pattern of local deformation and volume atrophy in both cortical (including the cingulate cortex, bilateral insula, precuneus, temporal, frontal and occipital regions, mid-posterior segment of the corpus callosum and right angular gyrus) and subcortical regions (involving the corona radiata, brainstem, basal ganglia, amygdala, thalamus and right hippocampus) when compared to iRBD patients without MCI or control subjects. These findings were correlated with diminished performance in visuospatial abilities, attention, and executive functions, and increased severity of motor symptoms [49].

Taking advantage of follow-up databases, several studies explored longitudinal changes in iRBD. Changes in cortical thickness were noticed in the bilateral superior parietal and precuneus, the right cuneus, the left occipital pole, and lateral orbitofrontal gyri compared to controls [47,64].

Also, some authors found brain patterns related to Dementia with Lewy Body (DLB) in patients with iRBD. More precisely, one study characterized DLB-related whole-brain cortical thickness spatial covariance pattern (DLB-pattern) from a group of DLB patients and repeated this measure during the follow-up in DLB, iRBD, and control groups, identifying phenoconverters to PD or DLB. The DLB-pattern was characterized by thinning of the temporal, orbitofrontal, and insular cortices and relative preservation of the precentral and inferior parietal cortices. In the iRBD group, the baseline DLB-pattern scores were significantly higher in dementia-first than in parkinsonism-first converters. In non-converters, their analysis showed that the DLB-cortical thickness pattern remained stable over time in the majority of the cases. However, individual patterns were heterogeneous [31].

A second study identified a latent variable linking atrophy in the basal ganglia, thalamus, amygdala, and frontotemporal grey and white matter, and expansion in the cerebrospinal fluid (CSF)-filled spaces, to impairment in measures of motor, cognitive, and autonomic functions. They used a Deformation-Based Morphometry (DBM) analysis and the deformation score from this specific pattern foretold the progression to and not PD [30]. Finally, in a last study, cortical thinning in frontal, occipital, and parietal areas was found in iRBD patients before they converted into PD or DLB compared to iRBD non-converters [47].

With regard to the basal ganglia and the brainstem, decreased local volume in the SN pars compacta (SNc) [65], caudate nucleus [22], lentiform nucleus [22], bilateral putamen [23], right thalamus [66], and abnormal surface contraction in the thalamus and lenticular nucleus [28] in iRBD.

Poorer quality of life, as measured by a high Parkinson’s Disease Questionnaire 39 items (PDQ-39) score, was associated with reduced normalized putamen volumes. Similarly, slower performance on the timed gait task was associated with reduced normalized caudate and putamen volumes [23]. 

In a study designed to investigate subcortical and cortical GM volume alterations associated with depressive and anxiety symptoms in iRBD patients [67], authors found GM volume loss in the caudate nuclei, left calcarine, and right cuneus compared to iRBD patients without significant depressive symptoms and controls. They also found GM volume loss in the left amygdala extending to the hippocampus in iRBD patients with clinically significant anxiety symptoms compared to iRBD patients without significant anxiety symptoms and controls. Moreover, higher severity of anxiety and depressive symptoms was correlated with lower GM volume in these regions in iRBD patients.

Rotating frame relaxation parameters (adiabatic T1ρ and T2ρ, and non-adiabatic RAFF4) tended to be higher in iRBD patients than in controls in the amygdala, hippocampus, and thalamus [68].

In some studies it was paradoxically an increase in the volume of GM of the frontal cortex, caudate [69,70], cerebellum [70] posterior lobe, putamen, and thalamus [24,69], hippocampi [50] interpreted as a possible compensatory mechanism without any concrete evidence to this day.

Finally, only four studies have found no significant clusters of reduced GM volume between iRBD patients and controls [41,43,71,72].

### 3.4. White Matter (WM) Integrity

The results of the microstructural WM assessment of iRBD were quite heterogeneous.

In the brainstem, decreases of Fractional Anisotropy (FA) in the SN [32], the tegmentum of the midbrain, and rostral pons associated with increases in Mean Diffusivity (MD) within the pontine reticular formation are reported [50,73]. One study found increased (but normal FA) in the SNc [74].

Mean free-water values (bi-tensor DTI) in the posterior SN for iRBD patients were reported to be higher and with a significant negative correlation with dopaminergic SPECT/CT activity [53]. On the contrary, in another study, MD was lower in SN pars reticulata (SNr) of iRBD vs. controls [68], but rotating frame relaxation parameters (adiabatic T1ρ and T2ρ, and non-adiabatic RAFF4) tended to be higher in iRBD patients than in controls in the midbrain and pons in favor of a degenerative process [68].

Further, significant decreases in the axial diffusivity were observed in the pons and the right SN, which were interpreted as altered axonal integrity [51].

At the supratentorial level, FA increases were found in the internal capsule bilaterally (along the anterior thalamic radiation), in the olfactory region, in right insula, right putamen [75] and corticospinal tract, cerebellar peduncles and brainstem [70] whereas significant FA decreases were observed in the fornix, the right visual stream (V1,V2) [51], the left post-central gyrus, the right lobule VIIB and left crus I of cerebellar hemisphere [75], the left superior temporal lobe [51,75] and corpus callosum [70].

Higher quantitative anisotropy in iRBD subjects compared with PD patients was found in bilateral middle cerebellar peduncles and right arcuate fasciculus thought as neural compensation [76] but more evidence is needed to support this hypothesis.

Increased MD in the iRBD compared to the PD group was found in the corpus callosum, the right limb of the internal and external capsule, the right superior and inferior longitudinal fasciculus, the right inferior frontal-occipital fasciculus, the right corticospinal tract [77], the right forceps major, the right corona radiata, the right tapetum, and the left posterior thalamic radiation [52].

Compared to controls, increased MD in the iRBD were found in left angular gyrus, left calcarine fissure and surrounding cortex, left and right caudate nucleus, left anterior and middle cingulate gyrus, left cuneus, right IFG pars orbitalis, left superior frontal gyrus medical orbital part, left fusiform gyrus, left hippocampus, left middle occipital gyrus, left rectus gyrus, left inferior and middle temporal gyrus, left thalamus, left lobule III of cerebellar hemisphere, left lobule VIIB of cerebellar hemisphere, and lobule X of vermis [75].

Radial diffusivity (RD) increases were observed in the fornix, the right visual stream (V1,V2), and the left superior temporal lobe [51]. In another study, the left caudate nucleus and right caudate nucleus had higher axial and radial diffusivity values [75].

These results were not consistent with other studies assessing axial, radial, or mean diffusivity and reporting no abnormalities [25,32]. 

Regarding structural connectivity data, several studies identified WM changes in iRBD. 

Using graph-theoretical analysis, a study demonstrated that individuals with iRBD exhibited notably increased small-worldness, clustering coefficient, and enhanced local connectivity among regions associated with olfactory, motor, and sleep functions when compared to control subjects. Betweenness centrality was also elevated in the left olfactory cortex, left precentral gyrus, left supramarginal gyrus, left temporal pole superior temporal part, lobule of X of vermis, and left crus II of cerebellar hemisphere [75], caudate nucleus and frontal cortex.

Moreover, with network-based statistics (NBS), a study showed greater connectivity between the right SMA area and right putamen, and between the left parahippocampal gyrus and left cerebellum in iRBD subjects compared to controls [78]. Here also, these results were interpreted in terms of neural compensatory mechanisms during the early stages of PD but without any strong evidence of this assessment.

Using probabilistic tractography, another study evaluated longitudinal degenerative progression patterns in prodromal PD patients (iRBD and/or hyposmia). A progression pattern was numerically quantified with a longitudinal brain connectome progression score. A trend towards two patterns of evolution of the connectome in prodromal PD patients (i.e., two subgroups) was found. Subgroup 1 (*n* = 11) (i.e., the subgroup that had the progression most similar to the PD group according to the longitudinal connectome scores) had increased and more variable Unified Parkinson’s Disease Rating Scale Part III Motor score (UPDRS-III) and Hoehn and Yahr scale (H&Y) scores at 1 and 1.5-year follow-up than at baseline. On the contrary, subgroup 2 (*n* = 5) (i.e., the subgroup that had the progression most similar to Controls) had UPDRS-III and H&Y scores near 0 at 1 and 1.5 year follow-up, similar to the scores achieved at baseline [79].

Using structural and functional neighborhood analyses, a study demonstrated that cortical thinning was constrained by the brain’s structural (and functional) connectome and that it mapped onto specific networks involved in motor and planning functions [21].

Further, in a third study using 7 Tesla MRI, was found impaired structural connectivity in 14 brainstem nuclei, including the connectivity between REM-on (for example, subcoeruleus nucleus) and REM sleep muscle atonia (for example, medullary reticular formation) areas [54].

The different methods of acquisition and post-processing of the signal are undoubtedly one of the reasons that can explain the differences observed between the reported studies. Further, as the prodromal phase of PD extends over many years, it is possible that there is some heterogeneity in the evaluated patients regarding their progression in neurodegeneration. Nevertheless, most of these studies showed that there was a microstructural alteration in the brainstem as well as in other brain regions. Efforts should be made to harmonize assessment techniques and obtain reliable markers.

### 3.5. Brain Iron Content

The most frequently evaluated feature in iRBD patients was the loss of dorso-nigral hyperintensity (DNH), mostly corresponding to nigrosome 1, in SWI sequences (Figure 3). However, this sign was inconsistent and therefore insufficient to make the diagnosis of the disease [37,38,55,56,57]. Indeed, even at ultra-high fields, 27.5% [56] to 61% [37,38] of iRBD subjects had DNH, in comparison with 96% of PD patients and 7.7% of healthy subjects. In iRBD patients who did not exhibit DNH abnormalities, there was a notably lower level of putamen dopaminergic SPECT/CT activity when compared to iRBD patients with DNH [37,56].

Evaluation of brain iron content using quantitative magnetic resonance imaging showed various results. On the one hand, no difference was found in the transverse relaxation rate (R2*) [32,72] in this region. On the other hand, several studies used Quantitative susceptibility mapping (QSM) to evaluate brain iron content in iRBD patients with different results. In the first one, QSM values in SN were higher in iRBD subjects than in controls (Figure 3) and were positively correlated with disease duration in the left SN [39]. In the second one, no difference was found in SN when comparing iRBD patients with healthy subjects two-years apart [58]. In the third one, in probabilistic regions of interest (ROI) of nigrosome 1, QSM values in iRBD patients were similar to healthy subjects [36]. Finally, a study found that iron content in the right caudate nucleus, right dentate nucleus, right dentate nucleus, and left caudate nucleus was higher and respectively correlated with visuospatial function, memory function, and alternate-Tap test. Also, iRBD patients had reduced volume of the right caudate nucleus in SWI [57].

QSM showed superior sensitivity for PD-related tissue changes in nigrostriatal dopaminergic pathways [80] but this remains to be demonstrated earlier in the degenerative process such as in iRBD patients.

### 3.6. Neuromelanin

All the studies found a concordant result, namely a reduction in the neuromelanin-sensitive volume and/or signal intensity in the whole SN [32,33,74,81] (but not in the SNc evaluated by its own [65,82]) or in the locus coeruleus/subcoeruleus complex [34,35,81] in iRBD compared to controls (Figure 4). Reduced signal in the locus coeruleus/subcoeruleus complex identified iRBD with 82.5% sensitivity and 81% specificity [35].

This anomaly appears to be the most consistent in the prodromal stage of PD.

Interestingly, two studies [83,84] found a reduced signal of neuromelanin of locus coeruleus in depression without iRBD (or undiagnosed) compared to controls. No difference was reported for SN. Nevertheless, late-onset depression is still classified as a moderate risk for prodromal PD with low sensitivity and specificity [3] because of the lack of dedicated studies.

### 3.7. Brainstem Metabolism

Only two studies were performed in iRBD in proton Magnetic resonance spectroscopy (MRS). A 3T MRI study showed a reduction of choline/creatine and N-acetylaspartate/Choline ratios in the pontine tegmentum [46] whereas a 1.5T MRI study reported no significant differences in choline/creatine, N-acetylaspartate/creatine, and myoinositol/creatine ratios between healthy subjects and patients [45]. Further investigations are warranted.

### 3.8. Functional Network Alterations

In iRBD, the most reported resting state fMRI (rsfMRI) result was a reduced nigro-striato-thalamo-pallidal inter- and intra-hemispheric connectivity [21,40,41,42,43,63,85,86,87,88].

This was especially the case for the connectivity of the caudate, putamen, and globus pallidus, bilaterally [41], as well as between the left SN and the left putamen [42] or between the brainstem and the cerebellum posterior lobe, temporal lobe, and anterior cingulate [63]. 

In comparison to controls, individuals with iRBD exhibited diminished cortico-cortical functional connectivity strength, particularly in edges located within posterior regions [89] but also they exhibited decreased nodal efficiency in the precentral gyrus, postcentral gyrus, supramarginal gyrus, superior temporal gyrus, supramotor area, straight gyrus, middle cingulate gyrus, and Rolandic operculum [88].

Finally, global functional dynamics and functional connectivity (FC) in iRBD were impaired within the resting-state network, particularly in the visual network and sensorimotor networks. FC parameters (synchrony, metastability, within- and between-network FC) were negatively associated with the clinical features, including UPDRS-I, II, Total, and RBD-Sleep Questionnaire [90]. 

Reduced FC between the brainstem and the cerebellum posterior lobe, temporal lobe, and anterior cingulate correlated with autonomic functions in iRBD (the Scales for Outcomes in Parkinson’s Disease-Autonomic scores) [63].

Comparing iRBD patients with and without MCI with controls, a study reported a lower FC strength between the left lateral occipital cortex and the nucleus basalis of Meynert (belonging to the cholinergic system) and the lingual gyrus in the iRBD group with MCI than in the iRBD with normal cognitive scores and healthy subjects. FC mean values positively correlated with the MoCA and memory test scores [91]. Another study found similar results in iRBD patients that were not considered as MCI: deficits in attention/working memory, executive function, and immediate memory were associated with abnormal striatal-cortical FC including frontal, temporal, and anterior cingulate cortices [87].

In patients with iRBD and impulse control disorders (ICDs), a functional hypoconnectivity between the limbic striatum and temporal occipital regions was also reported. Furthermore, the presence of ICDs correlates with hypoconnectivity between the limbic striatum and clusters located in the cuneus, lingual, and fusiform gyrus [43].

On the contrary, a study reported increased functional connectivity between the putamen and brain regions responsible for executing motion and coordination (such as the cerebellum, vermis, SMA, post- and precentral gyrus) and for planning movement (including the cuneus, precuneus, and superior medial frontal lobe) [44]. The authors interpreted this finding with compensatory mechanisms that enable smooth motion planning despite ongoing pathology. Further studies are needed to support this hypothesis.

Elevated connectivity between the right SN and the right cuneus/precuneus/superior occipital gyrus was reported [42] and increased functional connectivity between the left thalamus and occipital regions including the right cuneal cortex, left fusiform gyrus and lingual gyrus was positively correlated cognitive performance (word recognition list) suggesting a compensatory mechanism for cognitive impairment in iRBD [92]. This needs to be replicated to strengthen this hypothesis. In another study that examined olfaction in iRBD, researchers discovered elevated amplitude of low-frequency fluctuation (ALFF) values in the right parahippocampal gyrus. Additionally, changes in olfaction abilities were found to be associated with alterations in ALFF values in the occipital cortices [24]. 

Using the dynamic functional connectivity (dFC) approach, a study found four recurring dFC states that were characterized by significantly different connectivity patterns. iRBD patients spent more time in the frequent, fewer positive, and sparsely connected state (for which temporal properties were found to be associated with RBD screening questionnaire scores) than controls did while almost no time in the infrequent, more positive, and strongly connected state [93].

In an fMRI task involving bilateral hand movements, and in which there were internally selected and externally guided movement conditions for each hand, iRBD patients showed stronger functional recruitment compared to controls, mainly in the dorsolateral prefrontal, left primary somatosensory, the right fronto-insular cortices, in absence of behavioral differences. In these patients, lower MoCA scores were associated with higher activation in the left superior medial frontal and medial orbital gyrus, middle cingulate cortex, precuneus, and right cerebellum (Crus II) during the tasks [71].

Additionally, individuals with iRBD exhibited a decrease in BOLD signal change within the dorsal caudate nucleus and displayed notably distinct corticostriatal functional connectivity patterns compared to control subjects during a dual-task virtual reality gait paradigm that involved foot pedals [94].

Finally, in a virtual reality gait paradigm where individuals with iRBD were compared to controls, it was observed that iRBD patients exhibited diminished BOLD signal changes in the left posterior putamen and right mesencephalic locomotor region. Additionally, there was reduced FC between the frontoparietal network and the motor network when these patients navigated narrow versus wide doorways [95].

### 3.9. Other Interesting Features

In a retrospective study [62], of the 129 individuals diagnosed with iRBD patients from 1990 to 2019, 61 (47.3%) developed a clinically defined synucleinopathy during follow-up, namely 30 (49.2%) PD, 26 (42.6%) DLB, and 5 (8.2%) MSA. Of the five iRBD patients who developed MSA, MRI was abnormal in four subjects before phenoconversion (80%), and showed one of these signs: putaminal rim, cerebellar atrophy, middle cerebellar peduncle atrophy, middle cerebellar peduncle hyperintensity, pontine atrophy or cross-bun sign.

A study used arterial spin-labeled imaging to examine changes in cerebral blood flow (CBF) in individuals with iRBD, aiming to identify alterations in brain perfusion associated with the condition. The findings revealed that the iRBD group exhibited notably reduced CBF values in the right middle frontal gyrus, right inferior frontal gyrus, and right insula. It is important to know that the iRBD group had a lower MMSE score (but still normal, >26/30) than in controls [96].

More recently, another study found profound hypoperfusion and microvascular flow disturbances throughout the cortex in patients compared to controls. They applied dynamic susceptibility contrast MRI to characterize the microscopic distribution of cerebral blood flow. In iRBD patients, the microvascular flow disturbances were seen in cortical areas associated with visual processing, recognition, and language comprehension and were associated with impaired cognitive performance [59].

Moreover, iRBD patients had significantly higher enlarged perivascular spaces (EPVS) in centrum semiovale (CSO), basal ganglia, SN, and brainstem than healthy subjects and PD. Higher CSO-EPVS and SN-EPVS burdens were positively correlated with the severity of clinical symptoms in iRBD patients [97]. Also, three recent studies proposed diffusion tensor image analysis along the perivascular space (DTI-ALPS) as a non-invasive image-analysis method for the evaluation of glymphatic-system dysfunction. The authors found a lower value of the DTI-ALPS index in iRBD indicating a reduction in perivascular diffusivity. This means a reduced glymphatic clearance function [60,61,98].

### 3.10. Lateralization of Abnormal Findings

Lateralization of brain structures or connectivity impairment varied across the studies we examined. In cases of symptomatic RBD, such as post-pontine strokes, both left and right lesions have been documented [99]. For neurodegenerative conditions, a hypothesis could be formed that, due to highly lateralized connections, the spreading pathology might initially be confined mainly to the same side. Recent research consistently linked handedness to predicting the side of PD onset, with symptoms more often manifesting on the dominant side, ranging from 60% to 65%. One might suggest that the lateralized irregularities in the dominant (left) hemisphere could be attributed to the prevalence of right-handedness in the majority of participants in these studies [100].

However, a study indicated that cortical degeneration in PD differed between the cerebral hemispheres, with early left hemisphere involvement and late right hemisphere involvement, often associated with posterior cortical atrophy. This pattern was not contingent on handedness or affected by the predominant side of motor symptoms [101]. Also, another study found that WM integrity significantly differed in PD with dominant symptoms on the right side, while it remained unchanged in PD with dominant symptoms on the left side (LDP), suggesting that the LDP profile might be linked to a more favorable prognosis [102].

The cause of this predominantly left hemisphere atrophy across various neurodegenerative diseases remains uncertain, although there are multiple hypotheses involving genetics, lateralized vulnerability, and disease-specific factors [101].

In most RBD studies, handedness was often not mentioned. Incorporating this information into subsequent clinical and imaging investigations could assist in elucidating the presence and importance of initial microstructural/connectivity lateralization impairment.

## 4. Discussion

The increasing number of MRI studies assessing brain alterations in patients with prodromal PD symptoms demonstrates a strong interest of the scientific community in identifying early biomarkers able to monitor neurodegeneration.

For this review, the literature search was limited to PubMed (MEDLINE) and Google Scholar databases for several reasons. First, a huge number of references were found due to the wide scope of MRI methods considered (*n* = 5246 identified; *n* = 77 included), which was also the strength of this article. To reduce the risk of omitting a report, the search was conducted twice at 6-month intervals and the bibliography was carefully evaluated by all authors. Compared with previous review articles on this topic, we included many more eligible articles. Those reviews chose either to discuss neuroimaging findings in iRBD in general (including nuclear medicine or ultrasound techniques, drastically limiting the coverage of MRI anomalies) [5,103,104,105,106,107,108] or simply not to cover new MRI techniques available nowadays (such as neuromelanin or spectrometry) [109,110]. Second, MEDLINE and Google Scholar are among the most used databases worldwide. Based on 58 published systematic reviews, a study aimed to determine how to do efficient searches in systematic reviews. Authors reported that MEDLINE median recall (number of included references retrieved by one database divided by the number of included references retrieved by all databases together) was 82.9%, compared with EMBASE 87.3%, Web of Science 72.5%, and Google Scholar 38.0%. Combining MEDLINE and Google Scholar, this score increased to 94.6% [111], which is a very good score. Finally, we followed highly recommended guidelines to conduct a review such as PRISMA [7] and Pautasso’s ten rules [112]. Thus, we can assume that this review covered the field as extensively as possible and is of interest to researchers and clinicians who wish to understand both iRBD pathophysiology, MRI techniques, and findings in 2023.

Some biomarkers such as frontal cortical and basal ganglia atrophy, reduction of neuromelanin in the SN and locus coeruleus, or increase of iron in nigrosome 1 were the most consistent and could help clinicians in the early decision-making of a prodromal synucleinopathy-related disease diagnosis. However, these MRI features still need to be investigated in larger cohorts and to be more widely tested longitudinally. Nevertheless, we should keep in mind that performing any single of these studies is difficult because of the diagnostic complex procedures for iRBD (requiring video-polysomnography) and the scarcity of patients with iRBD visiting a doctor. It was therefore worthwhile to carry out a review of the literature to cross-check the results and extract the most consistent results. 

Another problem was the lack of reproducibility of some of the studies leading to inconsistent results. As an example, we can evoke specific technical shortcomings of some of the manuscripts that may have used now known to be methods fraught with lots of statistically significant artifacts—like lack of rigorous quality control for movement in many diffusion and BOLD imaging studies. Thus, results strongly depend on the decisions made in terms of MRI preprocessing although structural imaging assessed with T1-weighted sequences shows greater reproducibility [113].

The field has matured over the course of 22 years making a lot of the earlier studies less convincing in retrospect.

Finally, the quality of the clinical movement disorders evaluation of the participants in the studies was not consistent across the studies. Indeed, a lack of parkinsonism from a movement disorders clinician is far more convincing than from someone not so trained. Unfortunately, the background of those who performed the clinical evaluation was not systematically specified. 

Nevertheless, we were able to find a common set of results for most of the MRI methodologies, the most difficult was obviously for fMRI. Further research is warranted, with larger cohorts to highlight discrete but constant abnormalities.

Technological advances in terms of image acquisition and processing allow higher imaging resolution. As an example, the development of the ultra-high field (7 Tesla) MRI, now makes it possible to explore very small anatomical structures and to reveal previously undetectable anomalies. This will undoubtedly enable to establish a more accurate correlation between certain brain regions that were previously difficult to segment (such as the locus coeruleus, the different parts of SN, the nigrosomes, and numerous other small brainstem nuclei) and clinical symptoms. Optimizing parameters for image acquisition and processing will help reduce some of the heterogeneity between studies. Furthermore, at this stage, none of the described MRI parameters has been consistently assessed from the perspective of predicting which synucleinopathy the patient will develop (e.g., PD, DLB, or MSA). This information is of the utmost importance to best adapt the care management. 

In the years to come, the combination of different imaging modalities (such as PET/SPECT and MRI) will certainly be of great help to support the hypothesis of the presence of brain lesions (nigrostriatal degeneration, neuromelanin reduction, iron accumulation, microstructural and connectivity changes, atrophy or increased volume thanks to MRI) and of the type of neurodegeneration (for example, alpha-synuclein, tau, amyloid thanks to PET). 

Then, diagnostic algorithms that include clinical, molecular, and imaging biomarkers in addition to genetic biomarkers may have a huge impact by identifying patients most at risk for neurodegenerative disease, and by allowing them to undergo the most appropriate therapeutic trial.

## Figures and Tables

**Figure 1 brainsci-13-01398-f001:**
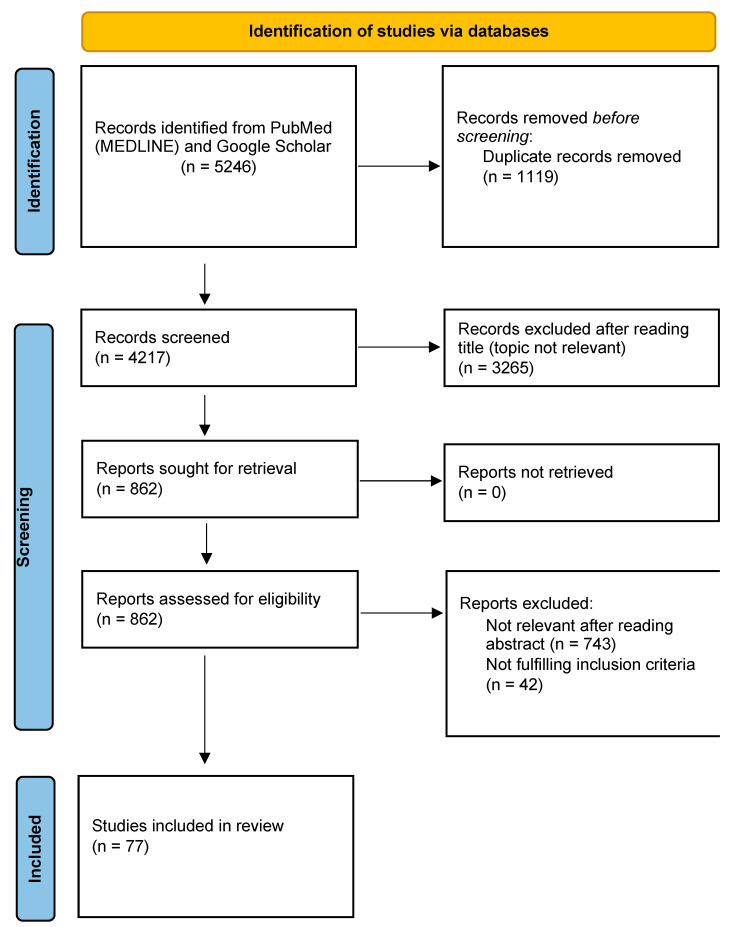
PRISMA flow diagram describing the steps of database search.

**Figure 2 brainsci-13-01398-f002:**
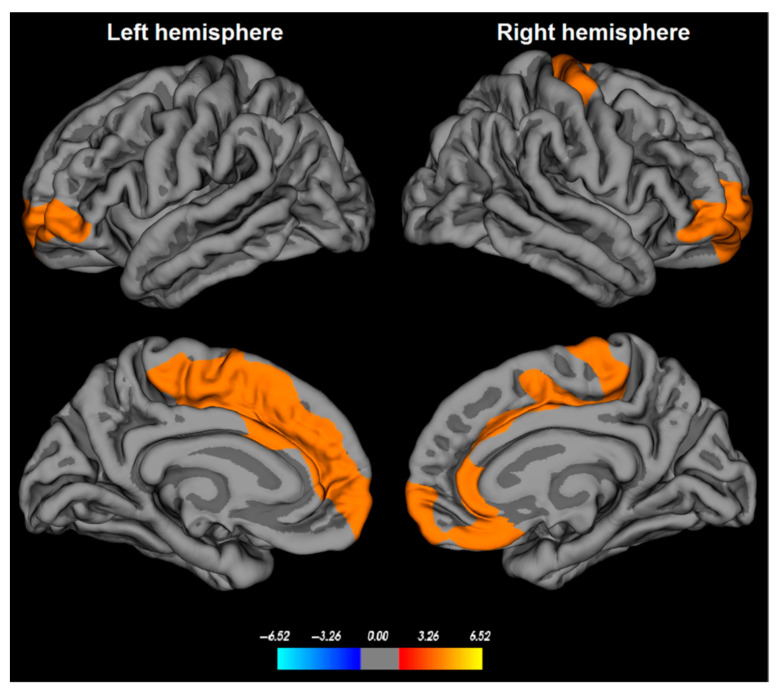
Cortical thinning in patients with idiopathic rapid eye movement (REM)-sleep behavior disorder (iRBD). Cortical thickness abnormalities in iRBD patients are found in bilateral medial superior frontal, orbitofrontal, and anterior cingulate cortices, as well as in the dorsolateral primary motor cortex in the right hemisphere. The color bar indicates the logarithmic scale of *p*-values (−log10) for between-group differences in cortical thickness, with red-yellow areas representing significant thinning in iRBD patients versus controls (corrected with a Monte-Carlo simulation using a *p*-value set at <0.05). From Rahayel et al., 2018 with permission [22].

**Figure 3 brainsci-13-01398-f003:**
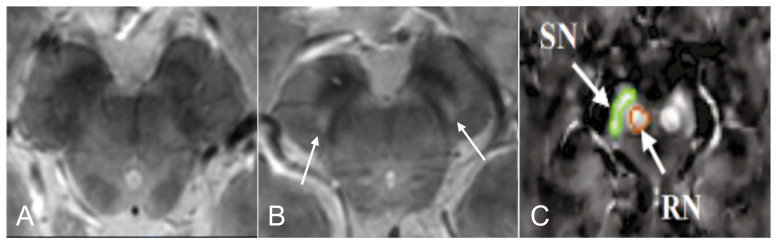
Brain iron content in idiopathic Rapid eye movement (REM)-sleep behavior disorder (iRBD). 3D GRE multi-echo susceptibility-weighted imaging MR of the substantia nigra (SN) in iRBD (**A**) and in healthy control (**B**). Nigrosome 1 (arrows) is visible bilaterally. Adapted from Frosini et al., 2017 with permission [38]. In (**C**) a quantitative susceptibility map (QSM), is seen in the SN (green) and the red nucleus (orange) in iRBD subjects (adapted from Sun et al., 2020 with permission [39]).

**Figure 4 brainsci-13-01398-f004:**
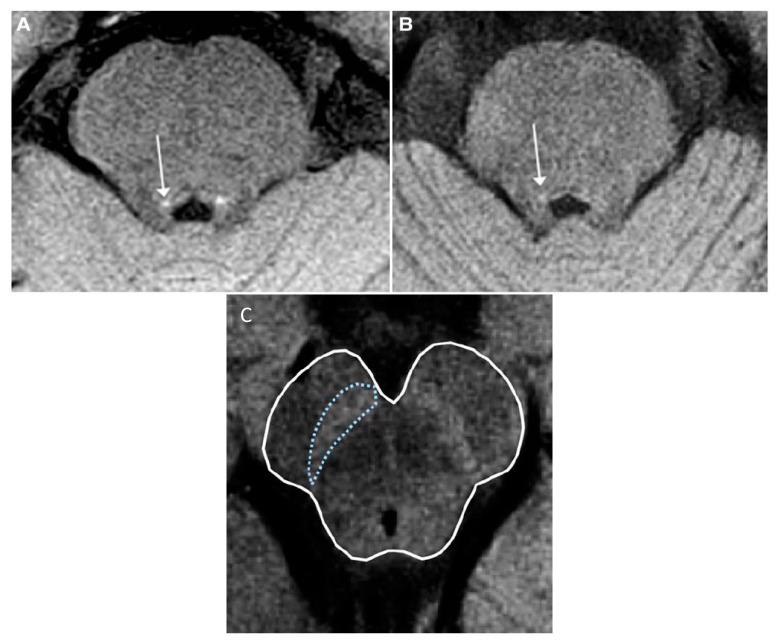
Neuromelanin-sensitive MRI in idiopathic Rapid eye movement (REM)-sleep behavior disorder (iRBD). Axial T_1_-weighted neuromelanin-sensitive images. (**A**,**B**): The coeruleus/subcoeruleus complex in a healthy volunteer (**A**) and a patient with iRBD (**B**). The locus area (arrows) is visible as an area of increased signal intensity. Adapted from Ehrminger et al., 2016 with permission [35]. (**C**): Neuromelanin in substantia nigra (outlined in blue) in a healthy subject. Adapted from Lehéricy et al., 2014 with permission [6].

## Data Availability

No new data were created or analyzed in this study. Data sharing is not applicable to this article.

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
