# Peer review of "Brain MRI Biomarkers in Isolated Rapid Eye Movement Sleep Behavior Disorder: Where Are We? A Systematic Review"

_brainsci, 2023, doi:10.3390/brainsci13101398_

Round 1

Reviewer 1 Report

GENERAL COMMENTS

The manuscript is a well-researched and comprehensively presented systematic review of MRI biomarkers in iRBD.

I believe the review will be more complete if one one additional issue is addressed:

    Asymmetry in clinical presentation, as well as in DAT scans, is usually present at the onset of PD motor symptoms. On the contrary, DLB and MSA usually do not show substantial asymmetries. It is, therefore, reasonable to hypothesize that MRI asymmetries should be expected in iRBD patients who will subsequently develop PD. Such asymmetries, if present, should be easy to demonstrate and might serve as useful diagnostic and prognostic markers. Remarkably, asymmetries are very infrequently quantitated in studies of iRBD and prodromal / early PD. In the current manuscript, “asymmetry” appears only in one reference (Zhong et al, 2019). It is likely, however, that the issue is raised in some MRI in iRBD studies but may not be prominently discussed. In any case, the manuscript may offer a novel and exciting insight into the value of MRI in iRBD if an additional effort is made to identify in the reviewed literature and discuss asymmetric MRI findings in iRBD, especially their prognostic value for development of PD.

    In some of the reviewed studies, certain findings refer specifically to “left” or “right” hemispheric structures. However, it is not discussed (at least in the current manuscript) whether these asymmetries have any relation to the subsequent appearance of PD and to its clinical lateralization.

SPECIFIC COMMENTS

Line 61

  Missing meaning of abbreviation NMC

  I presume it is “non-manifesting carriers”

Line 382, Figure 3

  Part C shows just a uniform gray background, whereas colors supposed to be present are missing

Author Response

Dear Reviewer,

I would like to express my sincere gratitude for your suggestion to enhance our literature review by advising us to address the question of lateralization of MRI abnormalities. This is indeed a very pertinent point, and as a result, we have added a dedicated paragraph to this issue. However, as you can imagine, providing a definitive answer to this question remains challenging.

Furthermore, we have corrected the abbreviation NMC, as you correctly identified.

Regarding Figure 3, it appears complete in our visualization, and the same seems to be the case for the other reviewer and the editor. We will make every effort to resolve any potential issues before publication if it is confirmed.

Once again, thank you for your valuable assistance.

Sincerely,

Paragraph added :

3.10. Lateralization of abnormal findings

         Lateralization of brain structures or connectivity impairment varied across the studies we examined. In cases of symptomatic RBD, such as post pontine strokes, both left and right lesions have been documented [99].For neurodegenerative conditions, a hypothesis could be formed that, due to highly lateralized connections, the spreading pathology might initially be confined mainly to the same side. Recent research consistently linked handedness to predicting the side of PD onset, with symptoms more often manifesting on the dominant side, ranging from 60% to 65%. One might suggest that the lateralized irregularities in the dominant (left) hemisphere could be attributed to the prevalence of right-handedness in the majority of participants in these studies [100].

However, a study indicated that cortical degeneration in PD differed between the cerebral hemispheres, with early left hemisphere involvement and late right hemisphere involvement, often associated with posterior cortical atrophy. This pattern was not contingent on handedness or affected by the predominant side of motor symptoms [101]. Also, another study found that WM integrity significantly differed in PD with dominant symptoms on the right side, while it remained unchanged in PD with dominant symptoms on the left side (LDP), suggesting that the LDP profile might be linked to a more favorable prognosis [102].

The cause of this predominantly left hemisphere atrophy across various neurodegenerative diseases remains uncertain, although there are multiple hypotheses involving genetics, lateralized vulnerability, and disease-specific factors [101].

In most RBD studies, handedness was often not mentioned. Incorporating this information into subsequent clinical and imaging investigations could assist in elucidating the presence and importance of initial microstructural/connectivity lateralization impairment.

Reviewer 2 Report

Abstract is well written. 

A systematic review paper cannot have a methodology and results section. It should identify appropriate sections to discuss the research domain with in-depth analysis. 

Introduction section is very weak. A good review paper should have a detailed discussion on the introduction, introducing the research domain in detail and should end with the summary of this research paper. 

Overall the paper looks like a summary of various other research papers. There is no in-depth analysis found. 

Table 1 is wordy. More table should be added to analyse the various literature

There should not be any bold sentences unless it is very important. 

Overall the organization of the paper needs a serious revision. I think the manuscript should undergo major revisions then it should be reviewed for technical quality. 

Minor editing of English language required

Author Response

Dear reviewer,

We sincerely appreciate your willingness to evaluate our manuscript, which required a significant amount of effort and energy to select 77 studies out of more than 5000 records identified. However, it is with some disappointment that we note your apparent dissatisfaction with the way we have structured the article. This stands in stark contrast to the other reviewer who commended our work, recognizing its potential value to our target audience, specifically physicians, and more particularly neurologists who may have limited knowledge, if any, about MRI in Parkinsonism and iRBD.

Regrettably, your feedback lacks specific details on areas for modification or improvement, leaving us to speculate about your concerns. Despite these challenges, we have made every effort to address your inquiries to the best of our abilities.

A systematic review paper cannot have a methodology and results section. It should identify appropriate sections to discuss the research domain with in-depth analysis.

We adhered to the author guidelines of Brain Sciences to align our manuscript with the journal's scope. Many high-quality reviews similarly adopt this framework, providing readers with a clear understanding of the research process, results, and a comprehensive discussion of the methods and findings.

It's important to note that our primary goal was not to conduct an in-depth analysis, as our intention was to create a systematic review rather than a meta-analysis. Instead, we aimed to consolidate all the findings within this highly specialized field and emphasize the most significant results in one cohesive document.

Introduction section is very weak. A good review paper should have a detailed discussion on the introduction, introducing the research domain in detail and should end with the summary of this research paper. 

As previously mentioned, we adhered to the author's guidelines, which state, "The introduction should briefly contextualize the study and emphasize its significance."

Given the nature of our paper as a review, we provide an in-depth exploration of the research domain in the results section, followed by a comprehensive discussion. The conclusion serves as a summary, and this information is also encapsulated in the abstract.

Overall the paper looks like a summary of various other research papers. There is no in-depth analysis found. 

Our primary objective was not to conduct an in-depth analysis in this review, as our aim was to create a systematic review rather than a meta-analysis. Instead, we gathered all the findings within this highly specific field and emphasized the most significant results. The discussion section delved into the variability of these results.

Table 1 is wordy. More table should be added to analyse the various literature

Table 1 has been revised for clarity. Regrettably, due to the absence of specific guidance, we had to make educated guesses regarding your expectations. To have a concise and clear summary of the information revealed by this literature review, we excluded the results from studies with moderate/low methodology and retained only those from the most robust studies. Thus, Table 1 contains fewer data, making it easier to read.

Originally, we included additional tables; however, we were advised to remove them because they were redundant with the text and did not contribute significantly to the article.

There should not be any bold sentences unless it is very important. 

Thank you for sharing this perspective. A previous reviewer actually suggested the opposite approach. They recommended using bold formatting to help readers easily distinguish the most significant findings from high-quality studies compared to smaller ones. However, if the editor deems it irrelevant, we are open to removing the bold font.

Overall the organization of the paper needs a serious revision. I think the manuscript should undergo major revisions then it should be reviewed for technical quality. 

We regret to come across this statement. Regrettably, without specific indications regarding which elements require improvement and the suggested modifications, we are unable to make the necessary revisions to this review. We are somewhat surprised by the lack of detailed feedback, which leaves us without clear guidance on how to enhance the manuscript, especially in contrast to the constructive feedback provided by the other reviewer. Nevertheless, we have endeavored to address the queries to the best of our ability.

Round 2

Reviewer 2 Report

The authors have clarified my doubts.